# Laboratory Study of Microsatellite Control Algorithms Performance for Active Space Debris Removal Using UAV Mock-Ups on a Planar Air-Bearing Test Bed

**Filipp Kozin** [1], **Mahdi Akhloumadi** [2] and **Danil Ivanov** [1,*]

[1] Keldysh Institute of Applied Mathematics RAS, Miusskaya sq. 4, 125047 Moscow, Russia
[2] Moscow Institute of Physics and Technology, State University, Institutsky Lane 9, Dolgoprudny, 141701 Moscow, Russia
* Correspondence: danilivanovs@gmail.com

**Abstract:** In this paper, a planar air-bearing test bed with unmanned aerial vehicles (UAV) was used to test a microsatellite motion control system. The UAV mock-ups were controlled by four ventilator actuators that imitated the satellite thrusters and provided the required acceleration vector in the horizontal plane, and torque along the vertical direction. The mock-ups moved almost without friction along the planar air-bearing test bed due to the air cushion between the test bed surface and the flat mock-up base. The motion of the mock-ups motion imitated the motion of satellites in the orbital plane. The problem of space debris can be solved using special microsatellite missions able to dock to space debris objects and change their orbit. In this paper, two control algorithms based on the virtual potentials approach and the State Dependent Ricatti Equation (SDRE) controller, were proposed for docking to a non-cooperative space debris object. The algorithms were tested in a laboratory facility, and the results are presented and analyzed, including their main features demonstrated during the laboratory study. It was shown that the SDRE-based control was faster, although the virtual potential-based control required less characteristic velocity.

**Keywords:** motion control; laboratory verification; space debris removal; microsatellite

## 1. Introduction

Before the launch of any spacecraft mission, all of its systems go through a series of laboratory tests to confirm their performance. In particular, motion control systems are tested in a laboratory facility using aerodynamic suspensions, which makes it possible to simulate the conditions of orbital flight in some ways. There are two common types of aerodynamic test bed: a three-axis motion simulation for attitude, and a translational motion simulation for planar movement. A planar air-bearing test bed allows the performance of a number of tests regarding relative motion control algorithms in satellite formation flying missions [1], and is also used to study the docking performance of different capturing systems including robotic manipulators [2]. Some mock-ups are able to move in five degrees of freedom as in [3], but most provide imitation of only three degrees of freedom relative to attitude and translational motion. Nevertheless, the simplified planar translational and single-axis attitude motion of satellite mock-ups helps developers to reveal the main algorithm features of implementation in hardware that are not evident during the numerical simulation of the controlled motion [4–6].

Cold-gas thrusters can be utilized for satellite relative motion control in orbit. During laboratory experiments on the planar air-bearing test bed, such a control system can be used as it is (as in [5,7,8]), although the magnitude of the thrust in the condition of the Earth's atmosphere condition is different. Moreover, the onboard tanks with compressed gas require safety conditions in the laboratory that could be difficult to organize in some cases, especially if the laboratory facility is intended for educational purposes. Another

approach is to imitate the thrusters by actuators based on the ventilators of an unmanned aerial vehicle, as in [9]. It is quite clear that the ventilator cannot work in orbit, but it provides the corresponding value thrust to the cold-gas thrusters. Simplified versions of the control algorithm for the planar motion case can be tested by using UAV mock-ups with such thruster imitators.

The important problem of space debris in near-Earth orbits requires the development of special missions for active space debris object removal. The current state-of-the art developments in this field are presented in reviews [10,11]. One of the approaches implies the use of microsatellites capable of capturing a space debris object and changing its orbit by using onboard propulsion. There are a number of such missions, such as RemoveDEBRIS that used a net and a harpoon to capture an object [12], ESA CleanSpace mission also considered a harpoon to catch a debris object [13], and e.Deorbit mission used a robotic arm to grasp the launch adapter ring of the ENVISAT, which was considered debris [14]. Some elements of close proximity operations for an active space debris removal mission can be tested in the laboratory environment. A docking control algorithm for a satellite with flexible appendages was tested in [6]. In [4], a path planning method of a robotic arm was verified at an air-bearing test bed. Visual relative navigation algorithms were experimentally studied in [15,16].

This paper is devoted to the experimental study of two control algorithms for close-range approach to the defined point of a non-cooperative space debris object. The algorithms were based on the State Dependent Riccati Equation (SDRE) and a method based on virtual potentials. The equations of relative motion were nonlinear. The equations of motion were linearized near the current state vector to implement SDRE control [17–19]. Papers [20,21] presented a comparative study between SDRE and LQR based algorithms, and the SDRE based algorithm showed its advantages in terms of fuel consumption, approach time and trajectory accuracy. Algorithms based on SDRE have been used to solve various problems such as position and attitude control of a single spacecraft [22], or the relative motion control for formation flying of satellites [20]. For the problem of capturing a space debris object, it is necessary to take into account the effect of kinematic coupling, when the relative motion is considered not only as the motion of centers of mass of two bodies, but as the motion between two specific points fixed in the body reference frames, as considered in [23]. In [24], the application of control, based on SDRE for this type of relative motion equations, was investigated. In [1], the influence of the parameters of the control system on the execution of the control algorithm based on SDRE of the relative motion during capturing was studied, taking into account the saturation of the reaction wheels and the deviation of the thrust vector from the center of mass of the satellite.

The method of virtual potentials is effective in solving the problems of nonlinear control. To control a dynamical system using this method, it is necessary to construct a virtual potential field. The control actions are based on the vector fields obtained as a gradient of an artificial potential function. Such a field can be selected using various mathematical functions. For example, one of the types of potential functions was proposed in [25] in the form of an inverse-square function, which was applied to control the motion of a mobile robot. The virtual potential field can be constructed using harmonic functions and Laplace equations [26,27], artificial gyroscopic forces [28], flow functions from hydrodynamics [29], or using exponential series [30–32]. The method of virtual potential fields is used in physics, chemistry and biology [30], and in other areas. This method has been effectively applied for robotic control, for example, for traffic control problems for vehicles [32], cylindrical robots [33], unmanned aerial vehicles [34–37], and unmanned ground vehicles. Thus, artificial potential fields are widely used to control various dynamical systems.

The structure of this paper is as follows. In Section 2, a laboratory facility description is provided and its architecture is described. In Section 3, a short problem statement and the motion equations are presented. In Section 4, the description of the control algorithms and their simplification for laboratory testing is provided. In Section 5, the results of the experiments and algorithms comparison are presented.

## 2. Laboratory Facility Description

The laboratory facility COSMOS was developed by SputniX Ltd. for the Keldysh Institute of Applied Mathematics (KIAM), of the Russian Academy of Sciences. Its main parts are presented in Figure 1. An air duct was located under the table surface, and excessive atmospheric air pressure was created using the industrial ventilator. The air flowed through the holes on the table surface, thus, creating an air cushion under the flat base of the mock-ups. The table surface consisted of two aluminum plates. The size of the table was 1.5 by 2 m. The mass of the test bed was about 200 kg.

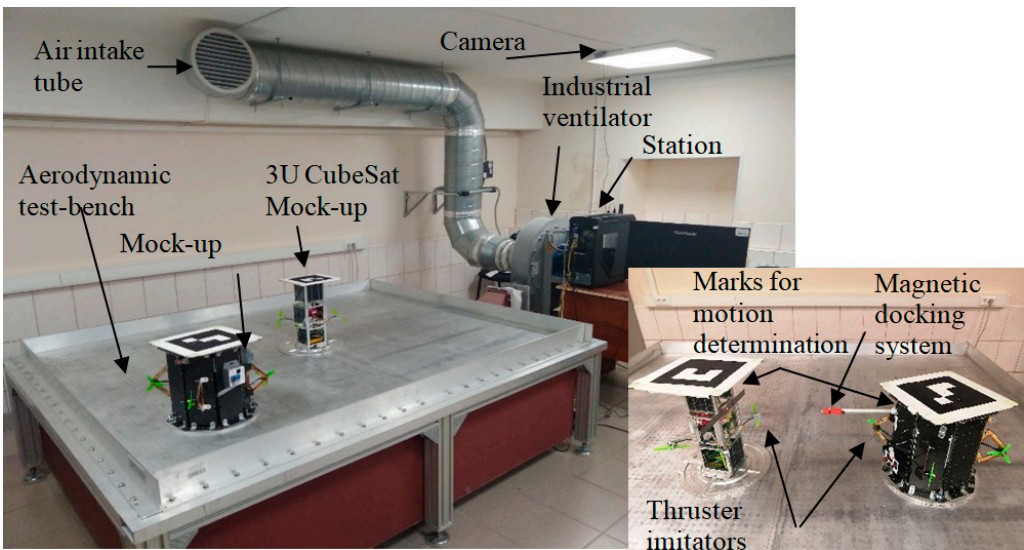

**Figure 1.** Planar air-bearing test bed located at KIAM.

The main advantage of such an aerodynamic test bed is that it provides the air cushion by itself. It is more convenient compared with other types of test beds with solid flat surfaces, where the air cushion must be produced by satellite mock-ups. However, the "air hockey"-like test beds are characterized by additional disturbances due to non-uniform airflow through the holes in the surface.

Satellites mock-ups were placed on the aerodynamic table, and Figure 1 shows two types of mock-ups. The mock-up control system was constructed using the hardware of the OrbiCraft constructor, developed by the SputniX company [38]. The satellite mock-up mass was 6 kg. The mock-ups consisted of the following systems:

- On-board computer *Raspberry PI B+*;
- Power system including the battery and PCU;
- Command transmitting system;
- A set of sensors for motion determination;
- Control actuators: one-axis reaction wheel, and four ventilators for thruster imitation;
- Passive magnetic docking system;
- *Wi-Fi* module.

The flowchart of all hardware component interactions is presented in Figure 2.

On the top of the mock-ups, ArUco markers [39] were placed for motion determination. Images from the observation camera above the table were processed for the mock-ups' centers of mass and attitude angle estimation. A desktop computer provided general experiment management, processed the observing camera images, transmitted the position measurements to the mock-ups, and logged the experiment data. Those data were transmitted to the mock-ups' onboard computers via Wi-Fi, and the measurements were processed by Kalman filter for the estimation of linear and angular velocity. Current state vectors were used for the reference trajectory calculation for each mock-up. The control commands for actuators were calculated according to the goal of the current task, taking

into account the disturbances acting on the mock-ups. The common control block-scheme is presented in Figure 3.

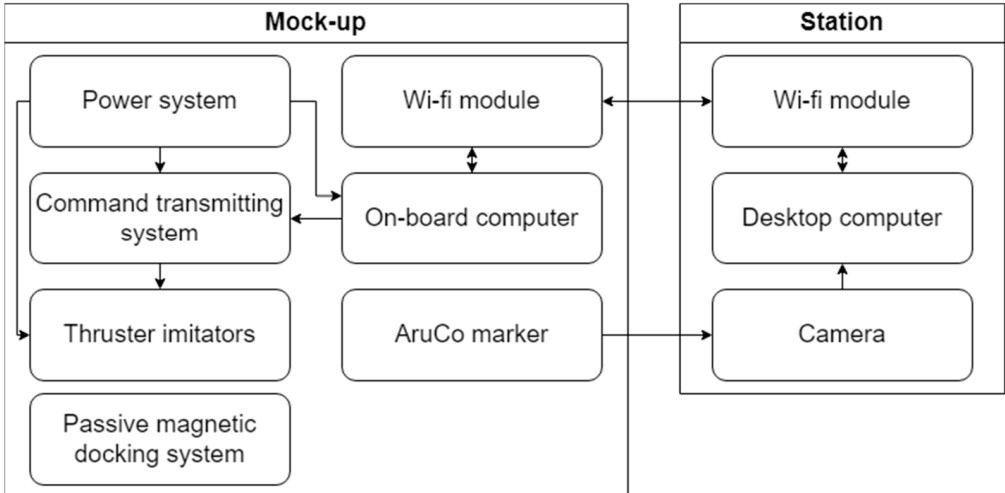

**Figure 2.** Flowchart of the mock-up and station hardware component interactions.

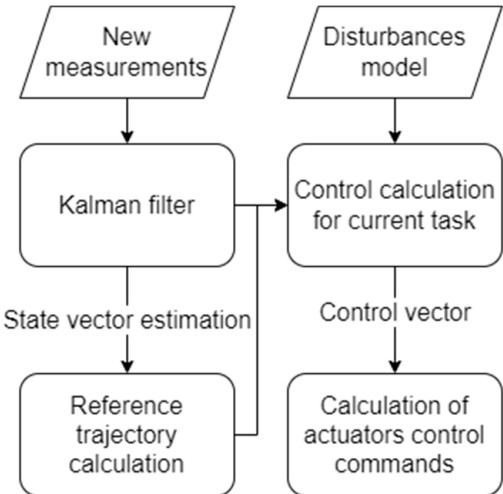

**Figure 3.** General block-scheme of the mock-ups control.

A special software was developed to run the required experiments in the laboratory facility. Its architecture allowed the addition of new blocks in the available experiment library, which met the testing experiment requirements. The software system consisted of a desktop program "*Station*", and "*Sat*" programs which were executed by the mock-ups' onboard computers. The experiment and test bed configurations were set by the input parameters from the config-file, the *Station* program connected to the *Sat* programs via *Wi-Fi*, and the *Sat* programs were initialized by the parameters from the experiment configuration. The *Sat* programs were launched by the *bash*-script from the desktop PC which then waited for connection to the *Station* program. The *Station* program was able to start and finish the experiment on the *Sat* program after successful connection. A general block-scheme of the *Station–Sat* interaction is presented in Figure 4.

In the next section, we consider the mock-up motion model used in the control algorithms for the problem of docking to the space-debris object.

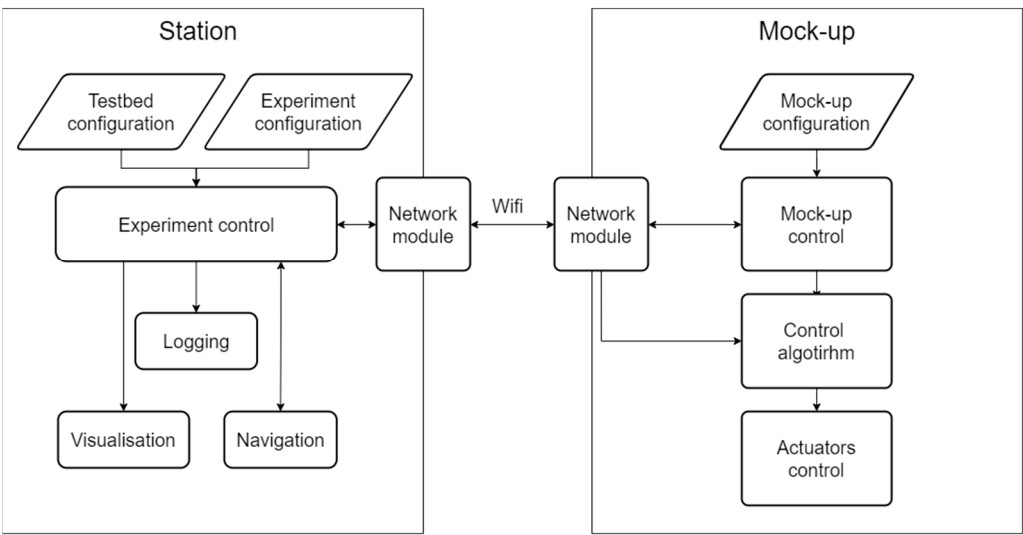

**Figure 4.** General scheme of the Station–Sat interaction.

### 3. Motion Model and Problem Statement

*3.1. Mock-up Motion Equations*

A planar air-bearing test bed only allows a partial imitation of the orbital flight conditions. Clohessy–Wiltchire equations [40] are often used as a relative motion model for satellite formation flying in near-circular orbits:

$$\begin{aligned}
\ddot{x} &= -2\dot{z}\omega, \\
\ddot{y} &= -y\omega^2, \\
\ddot{z} &= 2\dot{x}\omega + 3z\omega^2.
\end{aligned} \tag{1}$$

Here $\mathbf{r} = \mathbf{r}_2 - \mathbf{r}_1 = (x, y, z)$ is a radius vector of one of the satellites relative to the second written in the local-vertical-local-horizontal (LVLH) reference frame $OXYZ$, and $\omega$ is the orbital angular velocity. The origin $O$ of the LVLH is moving along the reference circular orbit, $OZ$ axis is aligned with the radius vector $\mathbf{R}_O$ of point $O$ from Earth center, $OY$ is aligned with the orbital angular momentum, and the $OX$ axis completes the right-handed triad (see Figure 5). These equations have the following solution:

$$\begin{aligned}
x(t) &= -3C_1\omega t + 2C_2\cos\omega t - 2C_3\sin\omega t + C_4, \\
y(t) &= C_5\sin\omega t + C_6\cos\omega t, \\
z(t) &= 2C_1 + C_2\sin\omega t + C_3\cos\omega t
\end{aligned} \tag{2}$$

where $C_1 - C_6$ are the constants dependent on the initial conditions. From (2) it can be noted that the motion along the $OY$ axis is independent and finite. In many formation flying control tasks, bounded relative trajectories are required to be achieved. Since the motion along the $OY$ axis is already bound, it may be omitted. Motion in the $OXZ$ plane corresponds to the motion in the orbital plane. Therefore, it is possible to imitate the orbital plane motion at the surface of the laboratory facility table.

The controlled motion equations in the orbital plane are as follows:

$$\begin{aligned}
\ddot{x} &= -2\dot{z}\omega + u_x, \\
\ddot{z} &= 2\dot{x}\omega + 3z\omega^2 + u_z,
\end{aligned} \tag{3}$$

where $\mathbf{u} = [u_x, u_z]^T$ is the vector of the control acceleration of the center of mass. For the tasks of docking with space debris, when the relative trajectory is close enough, the terms of non-inertial forces are negligible and can be compensated by the control algorithm, such as:

$$
\begin{aligned}
u_x &= 2\dot{z}\omega + \widetilde{u}_x, \\
u_z &= -2\dot{x}\omega - 3z\omega^2 + \widetilde{u}_z,
\end{aligned}
\tag{4}
$$

where $\widetilde{u}_x, \widetilde{u}_z$ are the control accelerations calculated according to the control algorithm.

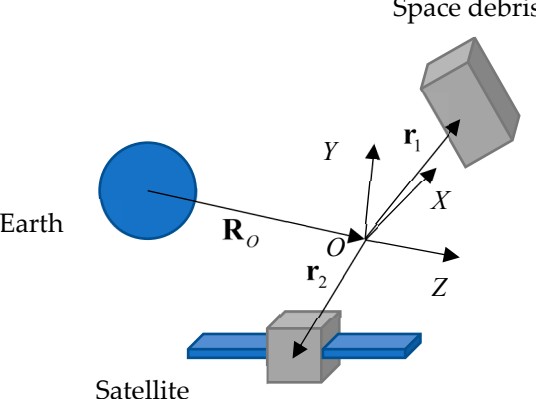

**Figure 5.** LVLH reference frame.

This results in the following mock-up motion control equations:

$$
\begin{aligned}
\ddot{x} &= \widetilde{u}_x, \\
\ddot{z} &= \widetilde{u}_z.
\end{aligned}
\tag{5}
$$

Attitude motion cannot be completely modeled in the laboratory. General angular motion is described by the Euler equations:

$$
\mathbf{J}\dot{\boldsymbol{\omega}} + \boldsymbol{\omega} \times \mathbf{J}\boldsymbol{\omega} = \mathbf{M},
\tag{6}
$$

where $\mathbf{J}$ is the inertia tensor, $\boldsymbol{\omega}$ is the angular velocity vector of the body reference frame relative to the inertial reference frame; $\mathbf{M}$ is the external torque. There are three degrees of freedom for the angular motion of a satellite in orbit. However, angular motion on the aerodynamic test bed has only one degree of freedom. Despite the limitations on orbital motion imitation, uniaxial angular motion makes it possible to test some simplified control algorithms.

Thereby, the motion equations of mock-ups on the test bed are:

$$
\begin{aligned}
m\ddot{\mathbf{R}} &= \mathbf{F} + \mathbf{F}_{dist}, \\
J\ddot{\alpha} &= M + M_{dist},
\end{aligned}
\tag{7}
$$

where $m$ is the mock-up mass; $\mathbf{R}$ is the radius vector of the center of mass in the reference frame fixed in the test bed; $\mathbf{F}$ is the control force acting upon the mock-up; $\mathbf{F}_{dist}$ is the disturbance force acting upon the mock-up; $J$ is the moment of inertia relative to the axis perpendicular to the plane of the test bed (vertical axis); $\alpha$ is the mock-up angle of rotation relative to the vertical axis; $M$ is the vertical component of the torque acting upon the mock-up; and $M_{dist}$ is the vertical component of the disturbance torque. There are two translational degrees of freedom and one angular. There are two types of disturbances: gravitational force defined by the test bed surface curvature, and aerodynamic forces due to the local heterogeneity of an airflow. It was assumed that the disturbance force and torque were negligible enough to be compensated by the thrusters' imitators.

### 3.2. Motion Equations for a Given Point of Satellite Relative to the Point of Space Debris

For the laboratory study of active debris-removal control algorithms, it was assumed that the satellite mock-up was equipped with a docking system capable of capturing the debris mock-up. Motion equations of the capturing system position relative to the capturing point of the space debris are required for the development of control algorithms. It was assumed that the capturing point was defined before the application of the docking algorithms.

Consider the relation between vectors of the $j$-th point at the surface of the space debris object with radius vector $\mathbf{r}_D^j$ (capturing point), and the $i$-th point of satellite $\mathbf{r}_C^i$ (capturing system position) (Figure 6):

$$\boldsymbol{\rho}_{ij} = \boldsymbol{\rho}_0 + \mathbf{r}_C^i - \mathbf{r}_D^j \tag{8}$$

where $\boldsymbol{\rho}_{ij}$ is the radius vector between two points in a debris-fixed reference frame, $\boldsymbol{\rho}_0 = [x; y; z]^T$ is the radius vector from the center of mass of the debris object to the satellite center of mass. The dynamical equations can be obtained as a result of second derivative calculation assuming that $\mathbf{r}_D^j = const$ in the debris reference frame:

$$\ddot{\boldsymbol{\rho}}_{ij} = \ddot{\boldsymbol{\rho}}_0 + \dot{\boldsymbol{\omega}} \times \mathbf{r}_C^i + \boldsymbol{\omega} \times (\boldsymbol{\omega} \times \mathbf{r}_C^i) \tag{9}$$

where $\boldsymbol{\omega} = \boldsymbol{\omega}_D - \boldsymbol{\omega}_C$ is the vector of relative angular velocity. In the case of the mock-up motion along the table surface, vector $\boldsymbol{\omega}$ has only one component $\boldsymbol{\omega} = \begin{bmatrix} 0 & 0 & \omega \end{bmatrix}^T$, and the equations for $\ddot{\boldsymbol{\rho}}_{ij}$ can be rewritten as follows:

$$\ddot{\boldsymbol{\rho}}_{ij} = \ddot{\boldsymbol{\rho}}_0 + \dot{\boldsymbol{\omega}} \times \mathbf{r}_C^i - \omega^2 \mathbf{r}_C^i \tag{10}$$

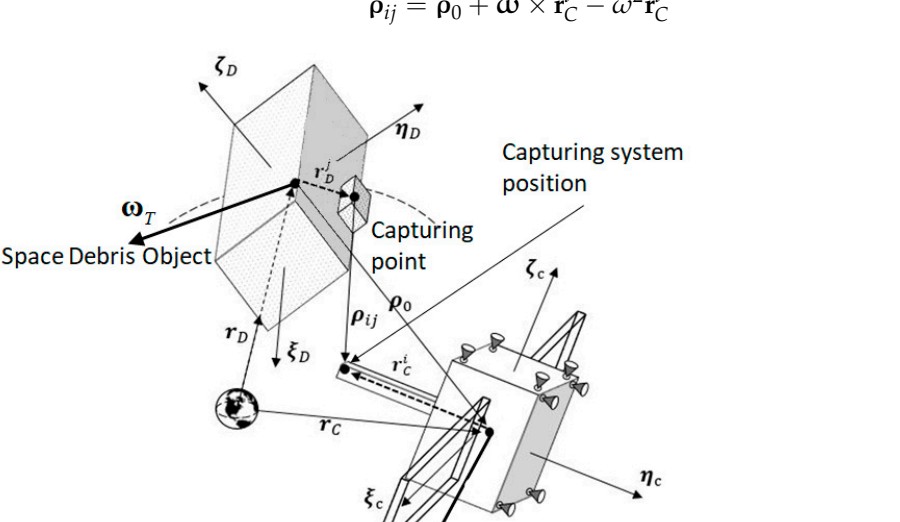

**Figure 6.** Reference frames for the space debris object removal task.

In this paper, it was assumed that the space debris mock-up motion was free, and the satellite mock-up was controlled. The center of mass translational motion relative to the mock-up center of mass can be obtained according to the equation $\ddot{\boldsymbol{\rho}}_0 = \mathbf{u}$, then the final relative motion equations between two points are as follows:

$$\ddot{\boldsymbol{\rho}}_{ij} = \mathbf{u} + \dot{\boldsymbol{\omega}} \times \mathbf{r}_C^i - \omega^2 \mathbf{r}_C^i \tag{11}$$

For the relative angular motion of two mock-ups, the equations are as follows:

$$J_D \dot{\omega} = J_D J_C^{-1} T_C \tag{12}$$

For the attitude control algorithms, it is convenient to use the unit vectors directed to the capturing system position $\mathbf{e}_C$ and to the capture point of space debris $\mathbf{e}_D$:

$$\mathbf{e}_C = \frac{\mathbf{r}_C^i}{|\mathbf{r}_C^i|}, \quad \mathbf{e}_D = \frac{\mathbf{r}_D^j}{\left|\mathbf{r}_D^j\right|} \quad , \tag{13}$$

where each vector is correspondingly written in each body reference frame. The relative vector between these vectors can be calculated as follows:

$$\mathbf{e} = \mathbf{D}\mathbf{e}_C + \mathbf{e}_D , \tag{14}$$

where $\mathbf{D}$ is the transition matrix from satellite to debris mock-up reference frame.

Using the second derivative of the vector $\mathbf{e}$, the following dynamical equations can be obtained:

$$\left(\frac{d^2\mathbf{e}}{dt^2}\right) = \left(\dot{\boldsymbol{\omega}}^\times + \boldsymbol{\omega}^\times\boldsymbol{\omega}^\times\right)(\mathbf{e} - \mathbf{e}_D), \tag{15}$$

where $\boldsymbol{\omega}^\times$ is a skew-symmetric matrix defined by the components of vector $\boldsymbol{\omega}$, and angular acceleration vector $\dot{\boldsymbol{\omega}}$ is defined by Equation (12).

### 3.3. Problem Statement

In this paper, the following orbital situation was modelled using space debris and active satellite mock-ups. It was assumed that the satellite was initially in the vicinity of the space debris object, i.e., the problem of far-range maneuvering was already solved. An active satellite is equipped with thrusters providing continuous actuation for the relative translational and angular motion, and these thrusters were imitated by the ventilators on the satellite mock-up. The orbital motion of a space debris object is not controlled, although its free motion on the aerodynamic table was modelled using the debris mock-up thrusters. All parameters of the satellite and the debris object were considered as known, and the relative motion was estimated by the on-board algorithms using the observation system data.

The satellite mock-up was equipped with a magnetic capturing system able to catch the space debris mock-up at the capturing point. It was assumed that the capturing system could be placed with a certain precision at the capturing point for successful docking. Moreover, the unit directional vector to the point of capturing system $\mathbf{e}_C$ was required to be antiparallel to the unit directional vector to the capturing point $\mathbf{e}_D$. The successful docking conditions can be written in terms of the center of mass position vector error $\Delta\boldsymbol{\rho} = \boldsymbol{\rho}_0 - \boldsymbol{\rho}_d$, where $\boldsymbol{\rho}_0$ is the current position vector between the centers of masses and $\boldsymbol{\rho}_d$ is the required vector, defined by the following:

$$\boldsymbol{\rho}_d^T = K \cdot \mathbf{e}_D^T, \quad K > \left|\mathbf{r}_C^i\right| + \left|\mathbf{r}_D^j\right| \tag{16}$$

In this paper, the docking was considered successful when the following conditions were satisfied:

$$|\Delta\boldsymbol{\rho}| < \varepsilon_\rho, \quad \left|\Delta\dot{\boldsymbol{\rho}}\right| < \varepsilon_v, \quad |\mathbf{e}| < \varepsilon_e, \quad |\dot{\mathbf{e}}| < \varepsilon_{\dot{e}} \tag{17}$$

where $\varepsilon_\rho, \varepsilon_v, \varepsilon_e, \varepsilon_{\dot{e}}$ are parameters determining an acceptable approach error.

The problem was stated as establishing a control for translational and angular motion of a satellite to provide successful docking conditions (17). For the simplified version of the algorithms, tests were performed on the air-bearing test bed with mock-ups of a space debris and an active satellite.

### 3.4. The Ventilator Actuators Thrust Model

The ventilator actuators were used to imitate the thrusters in the laboratory conditions. The force in the horizontal plane was generated using the ventilators, which allowed

the control translational acceleration of the mock-up center of mass to be established. Additionally, the actuators provided the control torque along the vertical axis. For simultaneous application of horizontal force vector and vertical torque, four ventilators were required. The selected mock-up ventilators' configuration layout is presented in Figure 7. The rotational direction of the ventilators was fixed in the body reference frame.

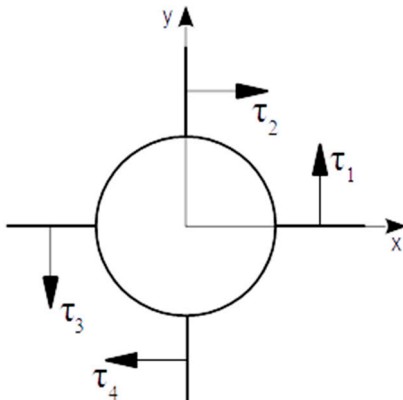

**Figure 7.** Mock-up ventilators layout in the mock-up reference frame.

In this paper, the following thrust model of the ventilator actuator was used:

$$F = \frac{1}{2} S \rho (V_e^2 - V^2) \tag{18}$$

where $V$ is the air velocity before entering the ventilator, $V_e$ is the air velocity after leaving the ventilator, $\rho$ is the air density, and $S$ is the area of the ventilator disc. The mock-up motion model under the action of a single ventilator can be presented by the following equations:

$$
\begin{aligned}
\ddot{x} &= \left(B - AV^2\right) \cos(\varphi - \varphi_0)/m, \\
\ddot{y} &= \left(B - AV^2\right) \sin(\varphi - \varphi_0)/m, \\
\ddot{\varphi} &= C\left(B - AV^2\right),
\end{aligned}
\tag{19}
$$

where $x$, $y$, $\varphi$ are the center of mass position coordinates and the attitude angle of the mock-up on the air-bearing test bed, $\varphi_0$ is the thrust vector attitude angle in the mock-up body reference frame, $A = S\rho/2$, $B = S\rho V_e^2/2$, $C = R/J$, $R$ is the moment–arm distance of the ventilator thrust, and $J$ is the inertia moment relative to the vertical axis. For this ventilator model, air velocity after leaving the ventilator was a function of the control $V_e = V_e(u)$, where $u$ is the control command. The calibration procedure was used to determine this dependency, and was implemented as a separate experiment.

When the mock-up was placed on the test bed at the initial state vector $\mathbf{q}_0 = \begin{bmatrix} x(t=0) & y(t=0) & \varphi(t=0) \end{bmatrix}^T$ and one of the actuators was turned on with a given control command $u$, the mock-up started to move according to the motion equations (19). Position and attitude of the mock-up on the test bed were monitored with the optical independent measurement system. The vector of the calibration parameters $\xi = [A, B, C]$ was different for every ventilator actuator. It was determined by the minimization of the following function:

$$\Phi(\xi) = \sum_{i=1}^{N} \left(\mathbf{q}_i - \tilde{\mathbf{q}}_i\right)^2 \tag{20}$$

where $\mathbf{q}_i$ is the state vector obtained with the independent measurement system, $\tilde{\mathbf{q}}_i$ is the state vector obtained with the motion equation integration, and $N$ is the number of measurement samples. With the minimization of (20), the dependency function $V_e = V_e(u)$ was estimated for a given value of $u$. To determine the full dependency function, it was necessary to conduct a series of experiments with different values of the control command $u$. In Figure 8, an example of the thrust value for every actuator of one of the mock-

ups is shown as a dependency on a dimensionless control parameter, which was varied from 0.55 to 1 in dimensionless units (it is a ratio of the applied voltage to the maximum available control voltage). The control parameter was the input value for the ventilator's control block, which was converted to voltage in the electrical thrusters' engines. In the case of low voltage, the ventilators could not start rotation. It was experimentally obtained that the minimum value of the control input was 0.55, at which the ventilators provided rotation. This dependence was approximated by the quadratic functions for onboard implementation.

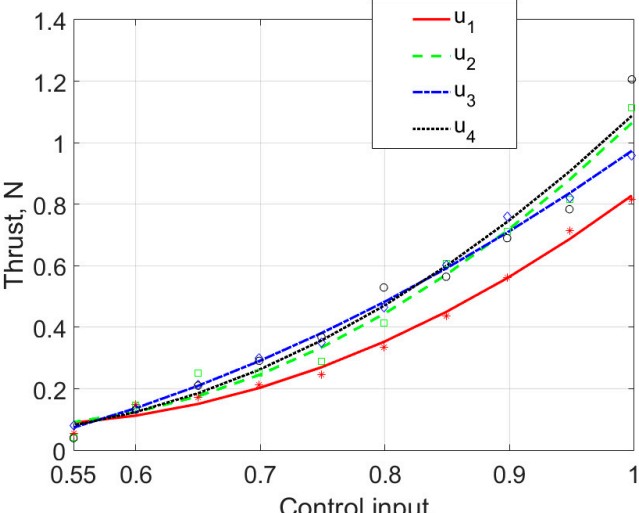

**Figure 8.** Thrust value for four mock-ups' ventilators depending on control command.

Using four ventilators, the required translational and angular control was implemented according to the conversion of the control vector *u* into the thrust of the ventilators. This conversion is described in [16].

## 4. Control algorithms for Mock-ups Rendezvous

The two control algorithms proposed in this paper aimed to achieve docking conditions (17). One of the algorithms was based on the SDRE control approach using the difference of the unit vectors from (14), and the other algorithm was based on the virtual potentials method for the control of the center of mass of the mock-ups. In this section, the basics of the control approaches are described and the application details of each algorithm are provided.

### 4.1. SDRE-based Control

SDRE control was applied to the nonlinear system in the following form:

$$\dot{\mathbf{x}} = \mathbf{f}(\mathbf{x}) + \mathbf{g}(\mathbf{x}, \mathbf{u}) \tag{21}$$

where $\mathbf{x}$ is the system state vector, $\mathbf{f}(\mathbf{x})$ is the nonlinear function of the system dynamics, $\mathbf{g}(\mathbf{x}, \mathbf{u})$ is the nonlinear control function, and $\mathbf{u}$ is the control vector. In the vicinity of the current state, the system can be linearized in the following form:

$$\dot{\mathbf{x}} = \mathbf{A}(\mathbf{x})\mathbf{x} + \mathbf{B}(\mathbf{x})\mathbf{u} \tag{22}$$

where $\mathbf{A}(\mathbf{x})$ and $\mathbf{B}(\mathbf{x})$ are the matrices of dynamics and control, respectively, which depend on the state vector. The control is formed as the closed-loop feedback control aiming for the following function minimization:

$$\widetilde{J} = \frac{1}{2} \int_0^{t_f} \left[ \mathbf{x}(t)^T \mathbf{Q} \, \mathbf{x}(t) + \mathbf{u}(t)^T \mathbf{R} \, \mathbf{u}(t) \right] dt \tag{23}$$

where $\mathbf{Q}, \ \mathbf{R}$ are the constant positive definite matrices, and $t_f$ is the fixed finite time. The control vector is calculated as follows:

$$\mathbf{u}(\mathbf{x}) = -\mathbf{R}^{-1} \left( \mathbf{B}^T(\mathbf{x}) \mathbf{P}(\mathbf{x}) \right) \mathbf{x} \tag{24}$$

where matrix $\mathbf{P}(\mathbf{x})$ is calculated as the solution of the following Riccati equation:

$$\mathbf{P}(\mathbf{x})\mathbf{A}(\mathbf{x}) + \mathbf{A}^T(\mathbf{x})\mathbf{P}(\mathbf{x}) - \mathbf{P}(\mathbf{x})\mathbf{B}(\mathbf{x})\mathbf{R}^{-1}\mathbf{B}^T(\mathbf{x})\mathbf{P}(\mathbf{x}) + \mathbf{Q} = 0 \tag{25}$$

For the problem of the mock-ups' rendezvous, the state vector is defined as follows:

$$\mathbf{x} = \begin{bmatrix} \mathbf{e} \\ \dot{\mathbf{e}} \\ \Delta\boldsymbol{\rho}_{ij} \\ \Delta\dot{\boldsymbol{\rho}}_{ij} \end{bmatrix} \tag{26}$$

where $\Delta\boldsymbol{\rho}_{ij} = \boldsymbol{\rho}_{ij} - \boldsymbol{\rho}_{ij}^d$, $\boldsymbol{\rho}_{ij}$ is the radius vector between the $j$-th point of the space debris mock-up (capturing point) and $i$-th point of the satellite mock-up (capturing system point), and $\boldsymbol{\rho}_{ij}^d$ is the required radius vector between these two points. In this paper, the vector $\boldsymbol{\rho}_{ij}^d$ was set to be constant in the debris mock-up reference frame, its value was determined by the acceptable error of the magnetic capturing system, and it was parallel to the unit vector of the capturing point:

$$\boldsymbol{\rho}_{ij}^d = K \cdot \mathbf{e}_D, \quad K > 0 \tag{27}$$

Taking into account the mock-ups' motion along the table surface, Equation (15) for vector $\mathbf{e}$ can be written in the following linearized form:

$$\begin{bmatrix} \dot{\mathbf{e}} \\ \ddot{\mathbf{e}} \end{bmatrix}_{4\times 1} = \begin{bmatrix} 0 & \mathbf{I}_{2\times 2} \\ \left( \dot{\omega}\widehat{\mathbf{J}}_{2\times 2} - \omega^2 \mathbf{I}_{2\times 2} \right) & 0 \end{bmatrix}_{4\times 4} \begin{bmatrix} \mathbf{e} \\ \dot{\mathbf{e}} \end{bmatrix}_{4\times 1} + \begin{bmatrix} 0_{2\times 1} \\ -\widehat{\mathbf{J}}_{2\times 2}\mathbf{e}_C^D \left( J_C \right)^{-1} \end{bmatrix}_{4\times 1} [T_C]_{1\times 1} \tag{28}$$

where $\mathbf{I}_{2\times 2}$ is identity matrix, and $\widehat{\mathbf{J}}_{2\times 2}$ is the symplectic matrix:

$$\widehat{\mathbf{J}} = \begin{bmatrix} 0 & -1 \\ 1 & 0 \end{bmatrix} \tag{29}$$

Assuming that the mock-up center of mass free motion is affected by small value disturbances, the approximation of the relative free motion can be considered as translational motion with constant velocity. In this case, the dynamical matrix of the system is as follows:

$$\mathbf{A} = \begin{bmatrix} \begin{bmatrix} 0_{2\times 2} & \mathbf{I}_{2\times 2} \\ \left( \dot{\omega}\widehat{\mathbf{J}}_{2\times 2} - \omega^2 \mathbf{I}_{2\times 2} \right) & 0_{2\times 2} \end{bmatrix} & 0_{4\times 4} \\ 0_{4\times 4} & \begin{bmatrix} 0_{2\times 2} & \mathbf{I}_{2\times 2} \\ 0_{2\times 2} & 0_{2\times 2} \end{bmatrix} \end{bmatrix}, \tag{30}$$

The control vector consists of the control torque $T$ and the control force vector $\mathbf{F}$ written in the debris mock-up reference frame:

$$\mathbf{u} = \begin{bmatrix} T \\ \mathbf{F} \end{bmatrix} \tag{31}$$

The control matrix, in this case, is as follows:

$$\mathbf{B} = \begin{bmatrix} 0_{2 \times 1} & 0_{2 \times 2} \\ -\widehat{\mathbf{J}}_{2 \times 2} \mathbf{e}_C^D (J_C)^{-1} & 0_{2 \times 2} \\ 0_{2 \times 1} & 0_{2 \times 2} \\ 0_{2 \times 1} & \mathbf{I}_{2 \times 2}/m \end{bmatrix}. \tag{32}$$

*4.2. Virtual Potentials-based Control*

The other control approach proposed in the paper was based on the virtual potentials method application. Its main idea was that the control of the satellite mock-up center of mass aimed to approach the mock-up space debris at a defined relative distance, and the satellite mock-up would move along the circle around the debris mock-up. In other words, the satellite mock-up would achieve the radial equilibrium position. Due to the debris mock-up angular motion, at some time moment, the satellite mock-up center of mass would achieve an acceptable position for docking to the debris mock-up capturing point. The acceptable position was defined by the sectorial area in the debris-mock-up reference frame. In this sector, the parameters of the virtual potentials were replaced by a different set of parameters in order to achieve the docking conditions.

The virtual potentials for the problem of docking were defined by exponential functions corresponding to the attractive and repulsive fields of central forces, $V_{\text{att}}$ and $V_{\text{rep}}$, of the following form:

$$\begin{aligned} V_{\text{rep}} &= C_{\text{rep}} e^{-\frac{\rho_0}{l_{\text{rep}}}}, \\ V_{\text{att}} &= -C_{\text{att}} e^{-\frac{\rho_0}{l_{\text{att}}}}, \\ V &= V_{\text{rep}} + V_{\text{att}} = C_{\text{rep}} e^{-\frac{\rho_0}{l_{\text{rep}}}} - C_{\text{att}} e^{-\frac{\rho_0}{l_{\text{att}}}}, \end{aligned} \tag{33}$$

where $C_{\text{att}}$ and $C_{\text{rep}}$ are positive attractive and repulsive parameters of the virtual fields of the forces, $l_{\text{att}}$ and $l_{\text{rep}}$ are the influence radiuses of the virtual fields, and $\rho_0$ is the distance between the mock-ups centers of masses. The center of mass motion equations under the application of the virtual potentials control were defined as follows:

$$\ddot{\boldsymbol{\rho}}_0 = -\text{grad}(V(\rho_0)) \tag{34}$$

This dynamical system was conservative, and the equilibrium position was defined by the following equation:

$$\text{grad}(V) = 0 \tag{35}$$

which led to the following:

$$\frac{C_{\text{att}}}{l_{\text{att}}} e^{-\frac{\rho_0^*}{l_{\text{att}}}} - \frac{C_{\text{rep}}}{l_{\text{rep}}} e^{-\frac{\rho_0^*}{l_{\text{rep}}}} = 0 \tag{36}$$

The stability of the radial equilibrium position $\rho_0^*$ is defined by the parameter values, which must satisfy the relation $l_{\text{att}} > l_{\text{rep}}$. For the asymptotic stability, a virtual radial friction $\mathbf{a}_{fr}$ should be applied to the system using the control actuators. The corresponding friction acceleration is defined by the formula:

$$\mathbf{a}_p = -f_r \dot{\rho}_r \mathbf{e}_r \tag{37}$$

where $f_r$ is the virtual friction in the radial direction, $\dot{\rho}_r$ is the velocity radial component, and $\mathbf{e}_r = \boldsymbol{\rho}_0 / \|\boldsymbol{\rho}_0\|$ is the unit vector along the radial direction.

The resulting satellite mock-up motion equations have the following form:

$$
\begin{aligned}
\ddot{x} &= \left( \frac{-C_{\text{att}}}{l_{\text{att}}} e^{-\frac{\rho_0}{l_{\text{att}}}} + \frac{C_{\text{rep}}}{l_{\text{rep}}} e^{-\frac{\rho_0}{l_{\text{rep}}}} \right) \frac{x}{\rho_0} - f_r \dot{\rho}_r \frac{x}{|\boldsymbol{\rho}_0|}, \\
\ddot{y} &= \left( \frac{-C_{\text{att}}}{l_{\text{att}}} e^{-\frac{\rho_0}{l_{\text{att}}}} + \frac{C_{\text{rep}}}{l_{\text{rep}}} e^{-\frac{\rho_0}{l_{\text{rep}}}} \right) \frac{y}{\rho_0} - f_r \dot{\rho}_r \frac{y}{|\boldsymbol{\rho}_0|}.
\end{aligned}
\tag{38}
$$

For docking, the satellite mock-up capturing system must always be oriented to the debris mock-up center of mass. In order to achieve this orientation, the SDRE-based attitude control was applied in terms of vector $\mathbf{e}_C$. After linearization, the dynamical motion equations were as follows:

$$
\frac{d}{dt} \begin{bmatrix} \Delta\mathbf{e}_C \\ \Delta\dot{\mathbf{e}}_C \end{bmatrix}^O = \begin{bmatrix} 0_{2\times2} & \mathbf{I}_{2\times2} \\ (-\omega_C^2 \mathbf{I}_{2\times2}) & 0_{2\times2} \end{bmatrix} \begin{bmatrix} \Delta\mathbf{e}_C \\ \Delta\dot{\mathbf{e}}_C \end{bmatrix}^O + \begin{bmatrix} 0_{2\times1} \\ \widehat{\mathbf{J}}_{2\times2} \mathbf{e}_C^D (J_C)^{-1} \end{bmatrix} \mathrm{T}_C
\tag{39}
$$

where

$$
\Delta\mathbf{e}_C = \boldsymbol{\Theta}_{CO} \mathbf{e}_C + \mathbf{e}_r
\tag{40}
$$

where $\mathbf{e}_r$ is a unit vector along the radius vector between the mock-ups' centers of masses, defined by:

$$
\mathbf{e}_r = -\frac{\boldsymbol{\rho}_0}{|\boldsymbol{\rho}_0|}
\tag{41}
$$

Two proposed control algorithms were experimentally studied on the mock-ups moving along the air-bearing test bed.

## 5. Results of the Laboratory Experiments

Due to the difference between the motion of the mock-ups along the surface and the satellite orbital motion, and due to the difference in the characteristics of the actuators, it is almost impossible to study real satellite control algorithm performance such as accuracy and time of docking. Nevertheless, the laboratory experiments revealed some features of the proposed algorithm scheme, and estimations on the range of algorithm parameters which provide successful docking.

During the experiments, the space debris mock-up tracked the defined trajectory along the test bed surface imitating free orbital motion on a circular orbit. The trajectory is described by the following:

$$
\begin{aligned}
x &= x_0 + A \cos(\omega t + \psi_0), \\
y &= y_0 + A \sin(\omega t + \psi_0), \\
\varphi &= \varphi_0 + \Omega t,
\end{aligned}
\tag{42}
$$

where $(x_0, y_0)$ are coordinates of the center of the orbit circle, $A$ is the radius of the orbit, $\omega$ is the angular velocity, $\psi_0$ is the initial phase angle, $\Omega$ is the constant angular velocity of the debris mock-up rotation around vertical, and $\varphi_0$ is the initial attitude angle of the mock-up. During the experiments, the following parameters were used:

$$
(x_0, y_0) = (0.8, 0) \text{ m}, \ A = 0.05 \text{ m}, \ \varphi_0 = \psi_0 = 0, \ \omega = \Omega = 6 \text{ deg/s}
\tag{43}
$$

The initial conditions for the translational and angular motion of the mock-ups were determined randomly, although the initial distance between the mock-ups was about 1 m. The algorithm control parameters are presented in Table 1.

**Table 1.** Control parameters.

| Control Parameter | Value |
|---|---|
| **Virtual potentials control** | |
| Virtual potentials parameters | $l_{\text{att}}, = 5,\; l_{\text{rep}} = 2.755,$ $C_{\text{att}} = 4.5,\; C_{\text{rep}} = 5,$ $f_r = 0.5$ |
| Virtual potentials parameters in the docking sector | $l_{\text{att}}, = 5,\; l_{\text{rep}} = 2.755,$ $C_{\text{att}} = 12,\; C_{\text{rep}} = 1,$ $f_r = 0$ |
| Matrix $\mathbf{Q}$ for the attitude control | $\mathbf{I}_{4\times4}$ |
| Matrix $\mathbf{R}$ for the attitude control | 0.1 |
| Docking sector size, deg | 20 |
| **SDRE-based control** | |
| Matrix $\mathbf{Q}$ | $\mathbf{I}_{8\times8}$ |
| Matrix $\mathbf{R}$ | $\begin{bmatrix} 0.1 & 0_{1\times2} \\ 0_{2\times1} & 10^4 \cdot \mathbf{I}_{2\times2} \end{bmatrix}$ |
| **Docking conditions** | |
| Acceptable position error, m | 0.05 |
| Acceptable attitude angle error, deg | 10 |

For safety reasons, when the satellite mock-up approached the debris mock-up in a manner not acceptable for docking conditions, the collision avoidance control was applied to the satellite mock-up. The dangerous distance was set in the experiments as 0.5 m. The collision avoidance control was applied in the direction along the radius vector from the debris mock-up center of mass to the satellite mock-up center of mass, with the aim of increasing the relative distance and avoiding dangerous proximity.

*5.1. Virtual Potentials-based Control*

Virtual potentials-based control provided the satellite mock-up trajectory with constant distance relative to the space debris mock-up, and the magnetic capturing mechanism of the satellite mock-up was oriented towards the debris mock-up center of mass. When the satellite mock-up entered the acceptable area for capturing, the parameters of the virtual potentials were replaced by another set, allowing the capturing system to approach the capturing point at the debris mock-up. The scheme of the experiment with algorithm implementation is shown in Figure 9.

Before the execution of the main part of the algorithm, all the parameters were initialized according to the experiment configuration obtained from the *Station* software. This included initialization of the motion equations parameters, Kalman filter parameters, and virtual potentials parameters; and all the initially obtained measurements were linked to the corresponding mock-ups. In the case that the initialization was performed and a new set of measurements was obtained, the algorithm main loop began to execute. Relative motion parameters were estimated by the onboard Kalman filter, and the current trajectory was calculated. The control was calculated according to the virtual potentials. If the docking conditions were not satisfied, the calculated control was implemented by the actuators. Otherwise, the experiment was over, as the docking was successful. If the algorithm failed to provide the docking, the mail loop was interrupted manually.

An example of the experiment results may be considered, and the video of the conducted experiment can be accessed in [41]. Figure 10 presents the satellite and debris mock-ups' center of masses trajectories in the table-fixed reference frame. The initial position of the satellite mock-up, equilibrium distance and docking point are shown. The relative distance time–history plot is presented in Figure 11. The equilibrium distance

in this experiment was set at 0.81 m. At about 53 s after the start of the experiment, the satellite mock-up entered the acceptable capturing area, and the virtual potentials was replaced by another set to provide the mock-up approaching the capturing point. The control provided the tracking of the unit vector of the capturing system direction $\mathbf{e}_C$ and the radius vector from the satellite mock-up center of mass to the debris center of mass, and the angle between these two vectors is presented in Figure 12. This angle was inside the acceptable area during the time of the whole experiment due to the control implementation.

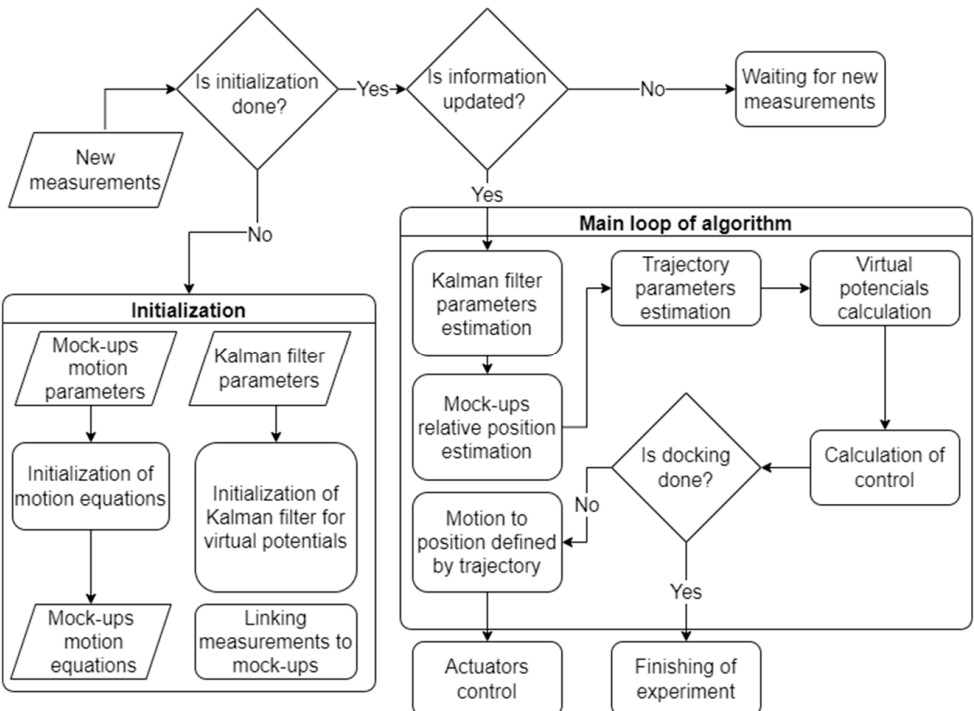

**Figure 9.** Scheme of the experiment with virtual potentials-based algorithm implementation.

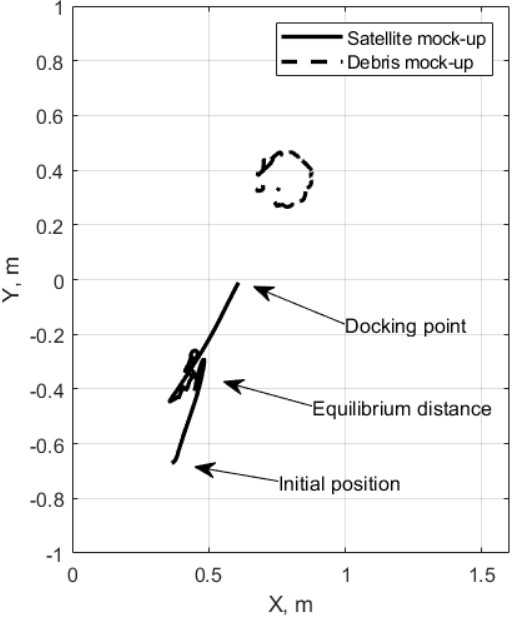

**Figure 10.** Satellite and debris mock-ups' center of mass trajectories in the table-fixed reference frame.

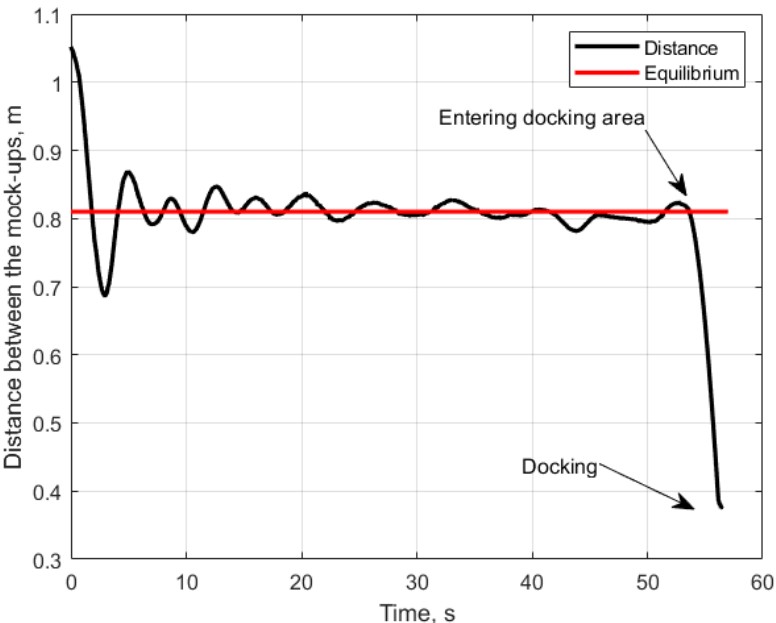

**Figure 11.** Relative distance of mock-ups' centers of mass.

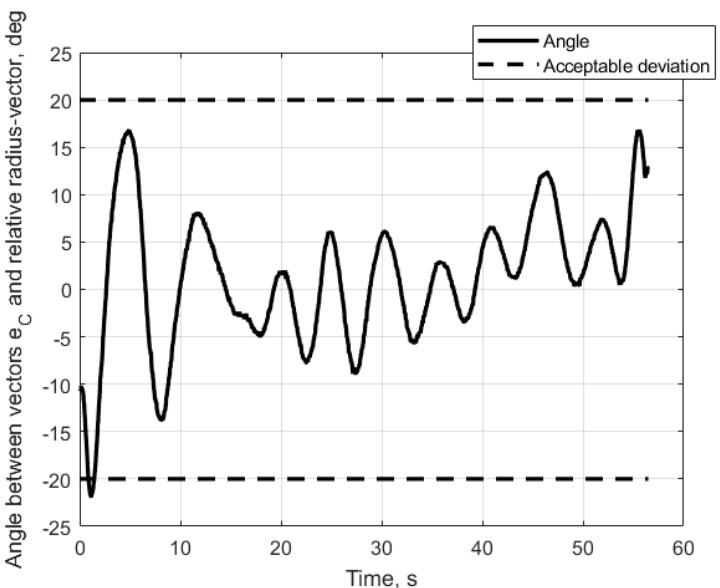

**Figure 12.** Angle between the capturing system direction and the radius vector from the satellite mock-up center of mass to the debris mock-up center of mass.

The calculated translational control acceleration according to the virtual potentials in (38) is presented in Figure 13. After the convergence to the equilibrium distance, the values of the control were low, and it compensated the disturbance forces acting on the satellite mock-up center of mass. At the point of entering the acceptable area, the values of the control became higher to provide the approach to the capturing point of the debris mock-up. In Figure 14, the angular control acceleration is presented, which is aimed at tracking the debris mock-up's center of mass by the direction of the capturing system. Note that the values of the angular control exhibited similar behavior as the deviation angle in Figure 12. This was due to the fact that the control aim was to set the angle to zero, while the disturbance torque caused some oscillations in this vicinity. The calculated translational and angular control was converted into the control commands of the thrusters; its values are provided in Figure 15. Note that according to the ventilators control implementation algorithm [16], at each instant, only three of the four ventilators were controlled.

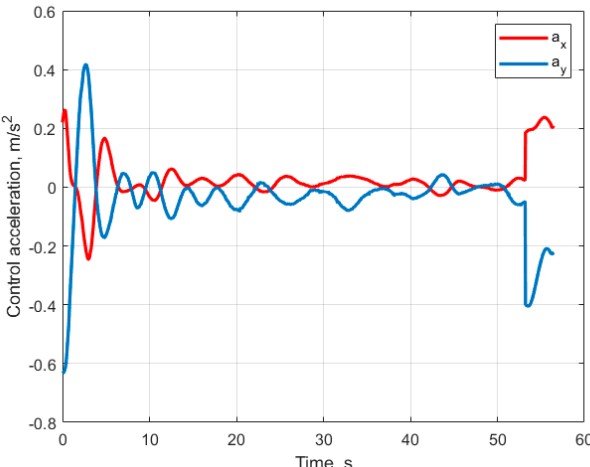

**Figure 13.** Virtual potentials-based control acceleration components.

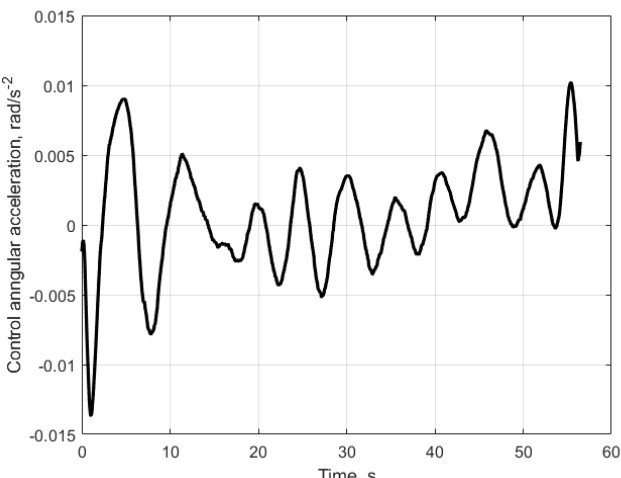

**Figure 14.** Angular control acceleration.

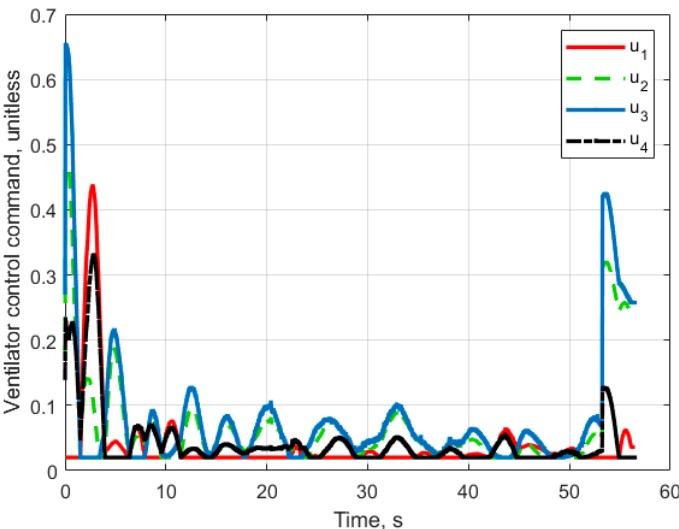

**Figure 15.** Mock-up thruster imitators' control values.

Figure 16 presents the angle of deviation between two unit vectors directed to the capturing system position $\mathbf{e}_C$ and the opposite vector of the capturing point of the space debris $-\mathbf{e}_D$. This angle represents the capturing conditions for the relative vector

$\mathbf{e} = D\mathbf{e}_C + \mathbf{e}_D$ explained in (17). It can be seen that due to the debris rotation, the value of this angle crossed the boundaries of acceptable error for docking. The acceptable area for docking capture is also defined by the angle between the unit vector $\mathbf{e}_D$ and the satellite mock-up radius vector in the debris body reference frame, as presented in Figure 17.

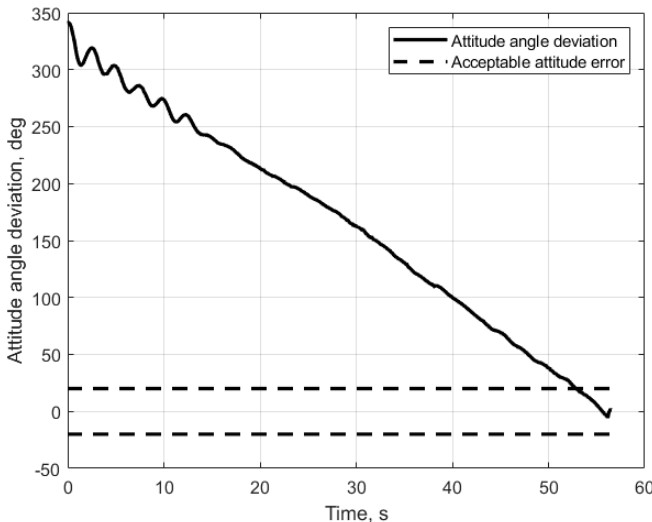

**Figure 16.** Angle between unit vectors directed to the capturing system position $\mathbf{e}_C$ and capture point of space debris $-\mathbf{e}_D$.

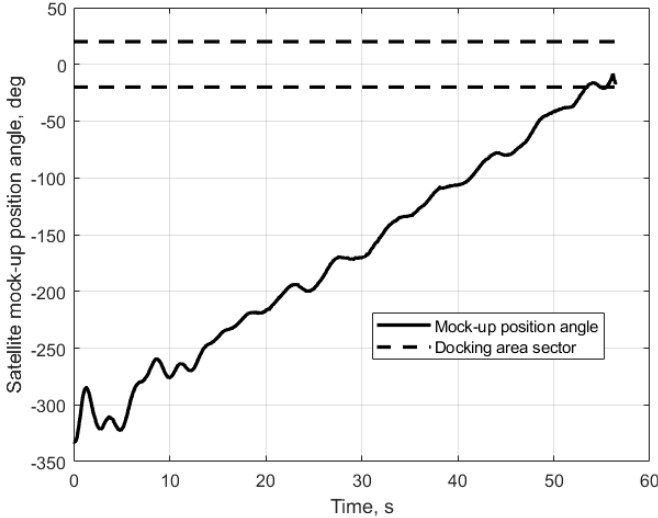

**Figure 17.** Angle between the unit vector $\mathbf{e}_D$ and the satellite mock-up radius vector in the debris body reference frame.

Thus, this experiment example demonstrated the proposed control scheme application for the mock-ups' rendezvous problem.

*5.2. SDRE-based Control*

A further experiment demonstrated the application of the proposed SDRE-based control. The video of the experiment is presented in [42]. The main difference of the control scheme was that the satellite mock-up aimed to achieve the center of mass and angular position required for docking, instead of waiting for the acceptable conditions at the equilibrium distance as in the case of virtual potentials-based control. The scheme of the experiment with SDRE-based algorithm implementation is shown in Figure 18.

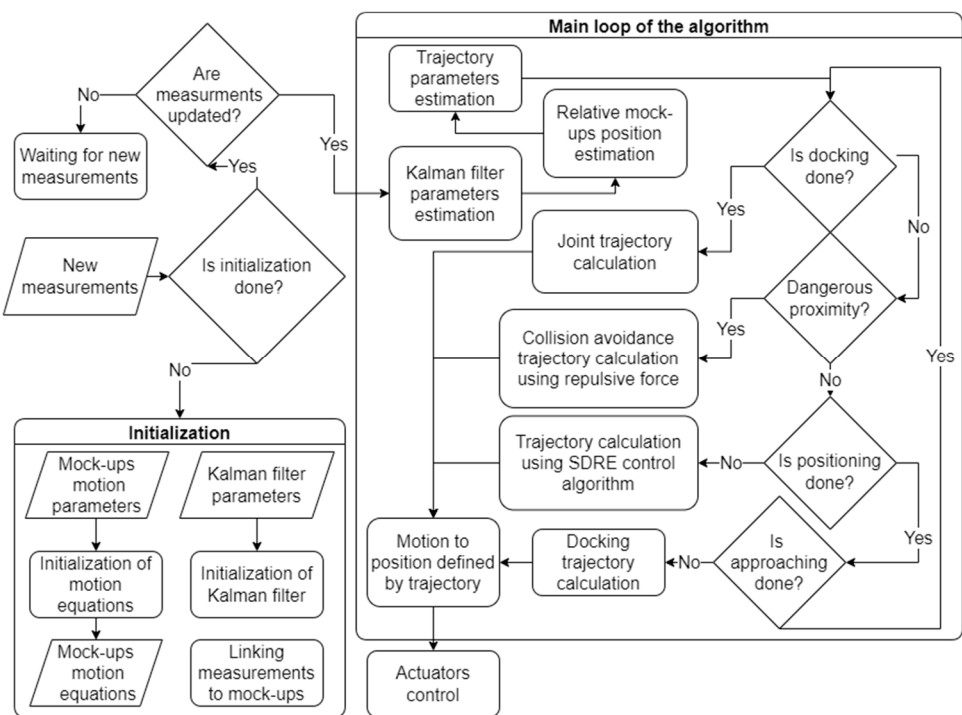

**Figure 18.** Scheme of the experiment with SDRE-based algorithm implementation.

Before the execution of the main part of the algorithm, all the parameters were initialized according to experiment configuration obtained from the *Station* software. This procedure was similar to the one from the virtual potentials control algorithm. If the initialization was performed, and a new set of measurements was obtained, the algorithm main loop began to execute. Relative motion parameters were estimated by the onboard Kalman filter. If the mock-up docking conditions were not satisfied, and there was no dangerous proximity between the satellite and debris mock-ups, the SDRE-based control was calculated and implemented by the thrusters' imitators. If the required relative position according to the SDRE algorithm was achieved, the satellite mock-up slowly approached the debris mock-up until the docking was successful. If the satellite mock-up was closer to the debris mock-up than the dangerous distance, and it was out of the docking sector, the collision avoidance control was applied in order to increase the relative distance. If the docking conditions were satisfied, the algorithm was switched to the joint trajectory motion mode, and it jointly stabilized the current position and attitude of both mock-ups. In the case that the algorithm failed to achieve docking, the main loop was interrupted manually.

Figure 19 presents the trajectories of the satellite and debris mock-ups in the aerodynamic table-fixed reference frame. The initial position and the achieved acceptable position for docking are shown in the plot. After the achievement of the required relative center of mass position and attitude, the approaching phase followed. During the approach, the required relative radius vector $\boldsymbol{\rho}_d$ length was slowly reduced until docking was achieved. The relative distance between the centers of masses of the mock-ups is presented in Figure 20. Due to the disturbances acting on the mock-ups and control implementation errors, the relative distance during the approaching phase reduced with some oscillations, although with linear trend, that led to the docking point. Figure 21 demonstrates the angle between the unit vectors directed to the capturing system position $\mathbf{e}_C$ and to the capture point of the space debris $-\mathbf{e}_D$ during SDRE-based control. It can be seen that the attitude acceptable for docking was almost achieved after 3 s, although the error periodically exceeded the limit value. The angle between the unit vector $\mathbf{e}_D$ and the satellite mock-up radius vector in the debris body reference frame is shown in Figure 22.

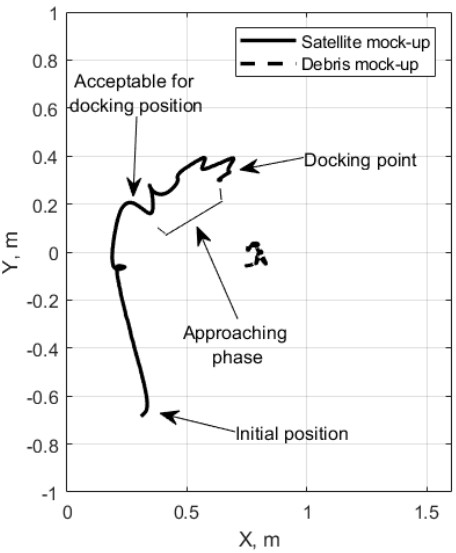

**Figure 19.** Center of mass trajectories of satellite and debris mock-ups in the table-fixed reference frame during SDRE-based control.

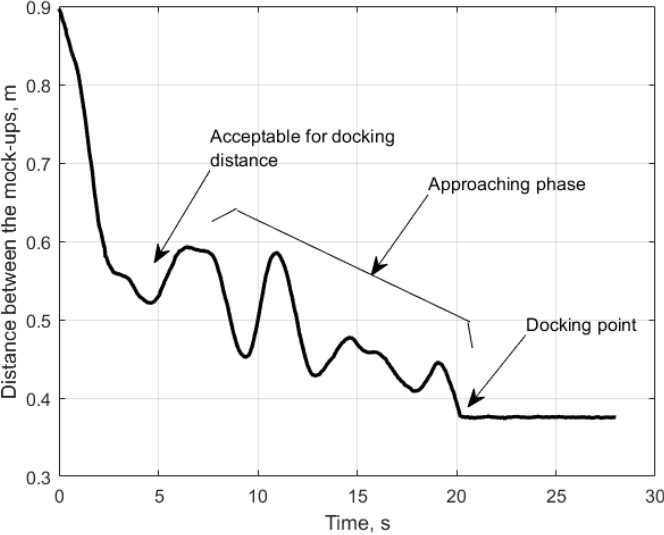

**Figure 20.** Relative distance of centers of masses of mock-ups during SDRE-based control.

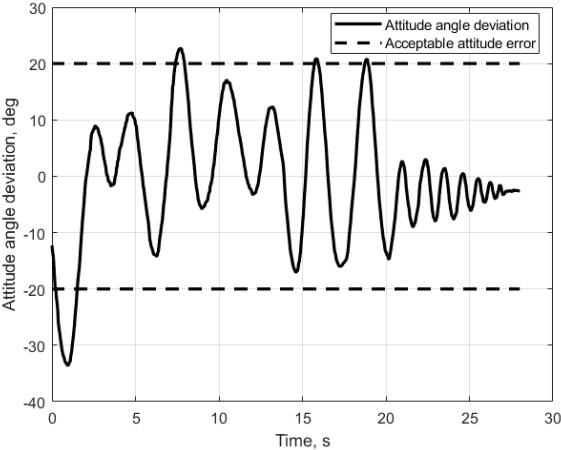

**Figure 21.** Angle between unit vectors directed to the capturing system position $\mathbf{e}_C$ and capture point of space debris $-\mathbf{e}_D$ during SDRE-based control.

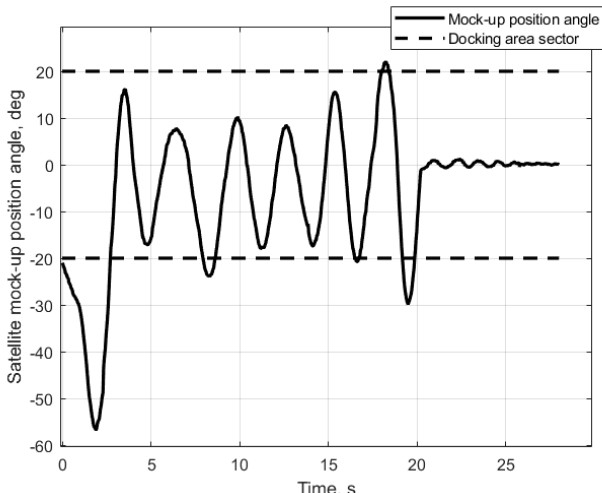

**Figure 22.** Angle between unit vector $\mathbf{e}_D$ and satellite mock-up radius vector in debris body reference frame during SDRE-based control.

### 5.3. Control Algorithms Experimental Study

The performance of the proposed control algorithms depended on a set of parameters. One of the most influential parameters was the angular velocity of the debris mock-up. A set of experiments with almost the same initial conditions for the satellite mock-up but with a different value of angular velocity of the debris mock-up was carried out. The results of the experiments were analyzed, the required $\Delta V$ for docking was calculated as the integral of the control acceleration components, and the docking time between the experiment start and successful docking was estimated. The results of the experiments are presented in Table 2. It was concluded that at an angular velocity lower than 12 deg/s, both algorithms successfully achieved docking, although at a value of 18 deg/s both algorithms failed to dock with the debris mock-up for different reasons.

**Table 2.** Results of the experiments with different debris mock-up angular velocities.

| Debris Angular Velocity, deg/s | Virtual Potentials | | SDRE | |
|---|---|---|---|---|
| | $\Delta V$, m/s | Docking Time, s | $\Delta V$, m/s | Docking Time, s |
| 3 | 6, 92 | 115 | 9, 07 | 20 |
| 6 | 6, 22 | 57 | 10, 35 | 20 |
| 12 | 6, 74 | 28 | 8, 77 | 23 |
| 18 | 17, 31 (fail) | 62 (fail) | 17, 69 (fail) | 14 (fail) |

In the case of successful docking with the same debris mock-up angular velocity, the virtual potential-based control required a lower value of $\Delta V$. The docking time was higher compared with the SDRE-based control because the virtual potential-based control simply waited for the acceptable relative position for docking, while the SDRE-based control sought to actively achieve this condition. The required $\Delta V$ for the virtual potential-based control was similar due to the low $\Delta V$ on the "waiting" distance, and differed for the SDRE-based control due to different active controlled motion time while achieving conditions acceptable for docking.

Examples of experiments with high angular velocity of the debris mock-up, when the algorithms failed to achieve docking, may be considered. Figure 23 shows the relative distance of the mock-ups' centers of masses during a fast debris rotation experiment with virtual potentials-based control. The satellite mock-up tried to dock three times during the experiment, but the time interval during which the docking area sector faced the docking satellite was too small to accomplish docking, as can be seen from Figures 24 and 25. This

time interval was about 2 s, the satellite mock-up center of mass approached the debris mock-up during this time. However, as the debris rotated, the docking satellite left the acceptable area and the virtual potentials parameters were switched back to obtain the equilibrium distance of 0.8 m. The control acceleration components are shown in Figure 26, where the peaks at the moments of entering and leaving the area of conditions acceptable for docking can be observed.

The case of SDRE-based control during the fast debris mock-up rotation experiment is presented in Figures 27–30. This experiment also did not achieve docking. As can be seen from Figures 27 and 28, the satellite mock-up entered a dangerous distance twice, that resulted in a collision avoidance control application (see Figure 30). The experiment was stopped after 15 s due to the fact that the satellite mock-up failed to move to the area acceptable for docking because of the fast debris rotation.

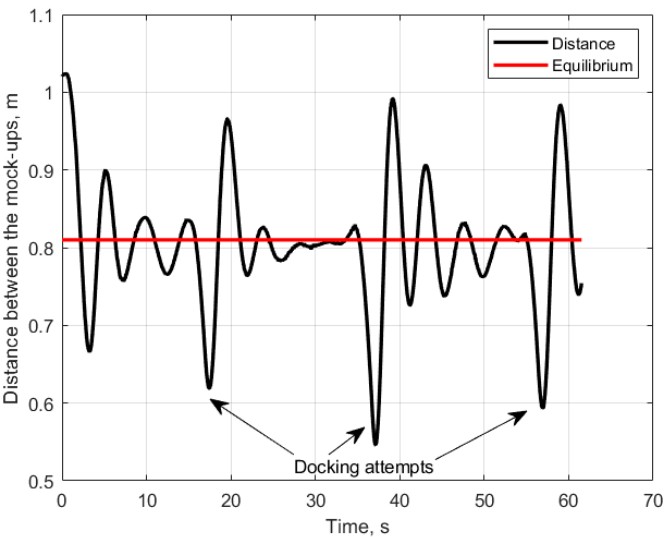

**Figure 23.** Relative distance of mock-ups' centers of mass during the fast debris rotation experiment with virtual potentials-based control.

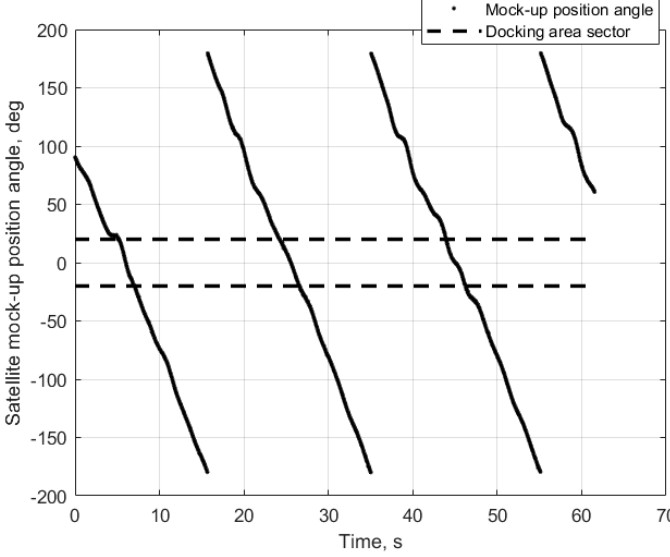

**Figure 24.** Angle between unit vector $\mathbf{e}_D$ and satellite mock-up radius vector in the debris body reference frame during the fast debris rotation experiment with virtual potentials-based control.

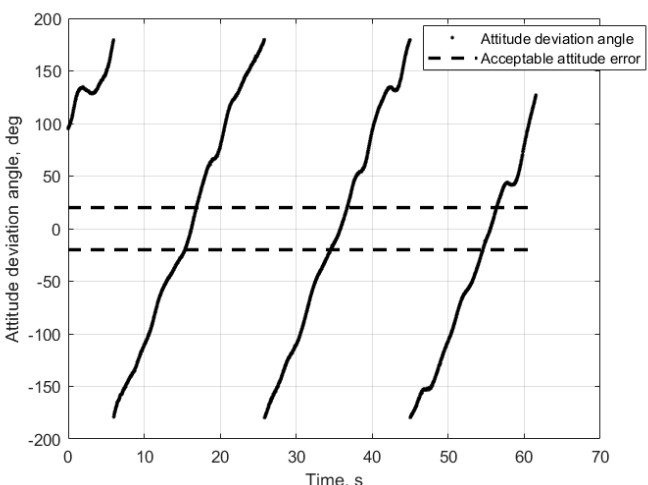

**Figure 25.** Angle between unit vectors directed to the capturing system position $\mathbf{e}_C$ and capture point of space debris $-\mathbf{e}_D$ during the fast debris rotation experiment with virtual potentials-based control.

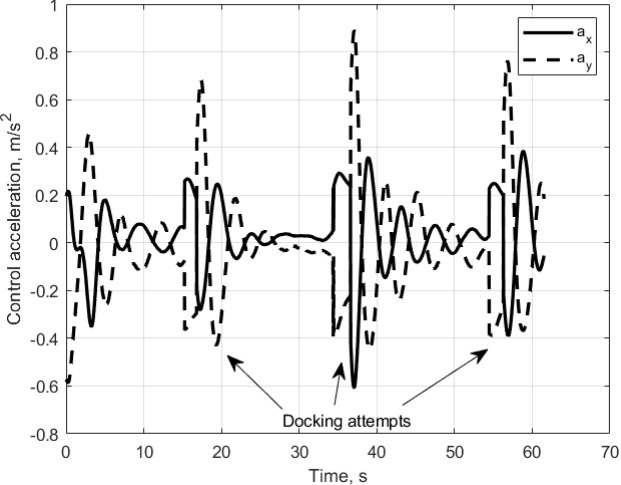

**Figure 26.** Virtual potentials-based control acceleration components during the fast debris rotation experiment with virtual potentials-based control.

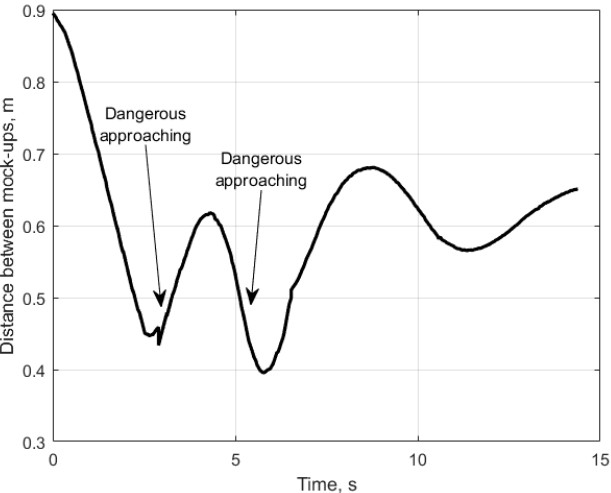

**Figure 27.** Relative distance of the mock-ups' centers of masses during the fast debris rotation experiment with SDRE-based control.

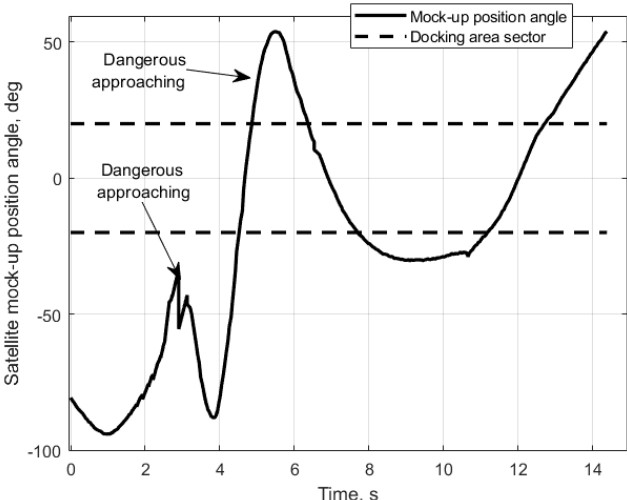

**Figure 28.** Angle between unit vector $\mathbf{e}_D$ and satellite mock-up radius vector in the debris body reference frame during the fast debris rotation experiment with SDRE-based control.

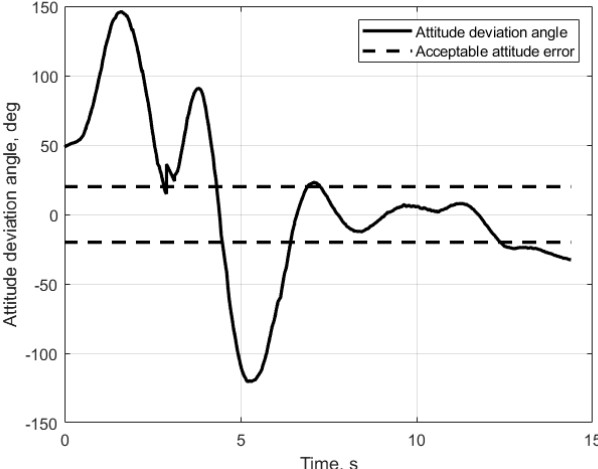

**Figure 29.** Angle between unit vectors directed to the capturing system position $\mathbf{e}_C$ and capture point of space debris $-\mathbf{e}_D$ during the fast debris rotation experiment with SDRE-based control.

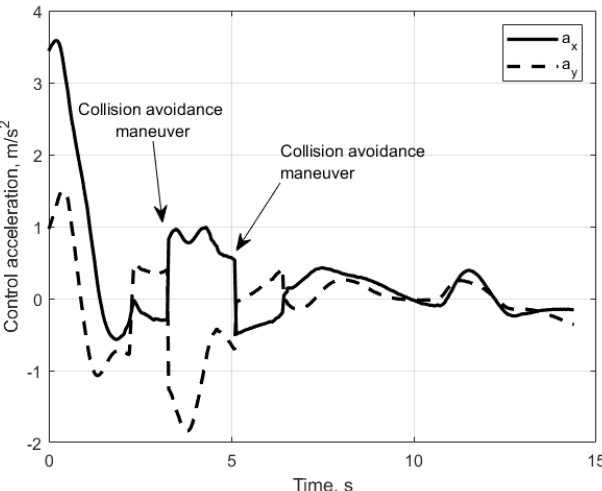

**Figure 30.** Virtual potentials-based control acceleration components during the fast debris rotation experiment with SDRE-based control.

Thus, the laboratory experiments showed the performance of the proposed algorithms and revealed their features using mock-up control systems. At high debris mock-up angular velocity, both algorithms were not able to achieve docking. At very low angular velocity, the virtual potential-based control waited for a considerable amount of time for the achievement of allowable docking conditions. In the case of orbital motion, the probability of entering the docking area may be low, depending on the debris angular velocity vector. The SDRE-based control required higher values of $\Delta V$, and could have led to a dangerous proximity, that required collision avoidance application.

## 6. Conclusions

Two algorithms for active satellite motion control relative to space debris were proposed to achieve successful docking, and were experimentally studied in this paper. The performance of the controlled motion of a satellite mock-up on the aerodynamic test bed is not the same as the performance of an orbital controlled motion, although the laboratory study allowed testing of the whole control logic, and revealed control scheme features such as principal dependence of the required $\Delta V$ and docking time on the debris mock-up angular velocity. As a result of this study, the two control algorithms can be recommended for onboard implementation in real active space debris removal missions with restrictions on the acceptable angular velocity of a space debris object for successful docking.

This work demonstrates that the different control approaches in space debris removal can be experimentally tested using the considered laboratory facility. Other types of control algorithms such as optimal control, model-predictive control, or some others, can also be adapted for mock-up application, and their performance can be tested. One of the directions for algorithm improvement will be to take into account the particular docking system features in the laboratory. The authors are planning to implement an autonomous visual-based navigation, test its performance, and estimate its influence on features of mock-up-controlled motion.

**Author Contributions:** Experimentation and software development, F.K.; motion equation and control algorithms development, M.A.; problem statement and study management, D.I. All authors have read and agreed to the published version of the manuscript.

**Funding:** This research received no external funding.

**Data Availability Statement:** Not applicable.

**Conflicts of Interest:** The authors declare no conflict of interest.

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
