# Peer review of "Laboratory Study of Microsatellite Control Algorithms Performance for Active Space Debris Removal Using UAV Mock-Ups on a Planar Air-Bearing Test Bed"

_drones, doi:10.3390/drones7010007_

Round 1

Reviewer 1 Report

Overall, the paper is well-defined throughout. The mathematics seems to be well-defined, and the results (including the plots and explanation) are very detailed. However, the structure and flow of the paper must be improved. Also, there are several typos and grammatical errors across the paper that must be corrected for better readability. Furthermore, it would be better to mention future work. Please find the comments below:

Section 3.1. Mock-up motion equations: It would be better to include a figure to better explain this section since the authors use the terminologies such as:  

“The origin O of the LVLH is moving along the reference circular orbit, OZ axis is aligned with the radius-vector RO of point O from Earth center, OY is aligned with the orbital angular momentum, OX axis completes the right-handed triad.” 

Having a figure that depicts the LVLH frame and OXYZ frame is better for the reader to understand this section. 

Line 351: The definition of the ‘E’ matrix is missing. I think it is the identity matrix and we usually use I instead of E. 

Line 38: “Nevertheless, simplified planar translational…” needs a reference.

Section 2: In this section, it would be better to include a figure/flowchart of all the components of the mock-ups since it is difficult to visualize the interaction between components without a flowchart. Or else, more definition of each component is required. 

Also, it would be good to see a comparison of the testbed used in this paper with other air-bearing testbeds.  

Highlight the advantages/disadvantages or special characteristics of this testbed since no prior work using this particular testbed is cited in the text.  

Line 313: The definition of the ‘dimensionless control parameter’ is missing. Also, why is the value of this parameter between 0.55 and 1?

Line 323: The section number is wrong. Should be Section 4 

Section 4. Results of the laboratory experiments: The flowcharts in this section require a more elaborate description/explanation. 

Reviewer 2 Report

 In the paper two control algorithms based on the vir-19 tual potentials approach and State Dependent Ricatti Equation (SDRE) controller are proposed for 20 the docking to the non-cooperative space debris object. The results of algorithms testing on the laboratory facility are presented and analyzed, their main features are demonstrated during the laboratory study. It is shown that the SDRE-based control is faster, thought the virtual potential based control requires less characteristic velocity.

As Fig.1 shown the test bed, please list some key parameters, the weight of cube sat and aerodynamic bench? How much forces of the thruster?

As shown in Fig.4, the capturing point of a noncooperative target is difficulty to determine, in case of test, how to identify? And how to simulate?

The first equation of Eq.2 is right?

Author Response

We thank the reviewer for the efforts in reading the manuscript. We addressed all the reviewer comments and suggestions. All the changes are highlighted in red in the manuscript.

Comment 1

As Fig.1 shown the test bed, please list some key parameters, the weight of cube sat and aerodynamic bench? How much forces of the thruster?

Answer:

We added in the text that the mass of the testbench is about 200 kg, the mass of the satellite mock-up is about 6 kg. The value of the thrust provided by the ventilators are presented in Fig. 8 – about 1 N each.

Comment 2

As shown in Fig.4, the capturing point of a noncooperative target is difficulty to determine, in case of test, how to identify? And how to simulate?

Answer:

We agree that in common case the determination of the capturing point is difficult task, but this task is out of scope of this work. In our work we assume that this point is already somehow defined. We added this comment in the text.

Comment 3

The first equation of Eq.2 is right?

Answer:

We checked the equation 2 and it seem to be correct.

Round 2

Reviewer 1 Report

Overall the readability of the manuscript was improved, but I found several grammatical mistakes and incomplete sentences. So, I recommend the authors carefully revise and submit your final manuscript for publication.